# A Comprehensive Analysis Examining the Role of Genetic Influences on Psychotropic Medication Response in Children

**DOI:** 10.3390/genes16091055

**Published:** 2025-09-08

**Authors:** Jatinder Singh, Athina Manginas, Georgina Wilkins, Paramala Santosh

**Affiliations:** 1Department of Child and Adolescent Psychiatry, Institute of Psychiatry, Psychology and Neuroscience, King’s College London, London SE5 8AE, UK; 2Centre for Interventional Paediatric Psychopharmacology and Rare Diseases (CIPPRD), South London and Maudsley NHS Foundation Trust, London SE5 8AZ, UK; 3Centre for Interventional Paediatric Psychopharmacology (CIPP) Rett Centre, South London and Maudsley NHS Foundation Trust, London SE5 8AZ, UK

**Keywords:** child, pharmacogenetics, mental health, thematic analysis, psychotropics

## Abstract

Psychotropic medication is commonly used for the treatment of mental health conditions. However, the genetic factors that influence psychotropic medication responses in children have not been thoroughly investigated. To address this gap, a systematic review and thematic analysis were conducted to examine the genetic impact of psychotropic medication response in children. The Down and Blacks and Consolidated Health Economic Evaluation Reporting Standards (CHEERS) checklists assessed the quality of studies and health economics, respectively. Using PRISMA reporting guidelines, 50 articles were identified with a sample size ranging from 2 to 2.9 million individuals. Most of the studies reported on ethnicity, and approximately half of the studies (24/50) were performed in North America. Five themes emerged from the thematic analysis: (1) implications of non-*CYP450* polymorphisms, (2) paediatric CYP450 pharmacogenetics, (3) genetic predictors of response, (4) insights for implementation and future research and (5) phenoconversion. The thematic analysis revealed that assessment of non-*CYP450* polymorphisms and psychotropic medication response, especially in those with mental health conditions such as autism, would be helpful. Epilepsy onset, risk and treatment response were associated with non-*CYP450* genetic variants. Phenoconversion of substrates associated with CYP2D6 and CYP2C19 metabolisers is common in individuals with mental health conditions, and *ABCB1* variants can influence psychotropic medication responses. A multidisciplinary model could also help guide clinical decision-making in cases involving complex neurodevelopmental profiles. Using the Down and Blacks checklist, the average score from the 50 studies was 17.7 points (min. 14, max. 24). The health economic evaluation of studies using the CHEERS checklist gave an average score of 33.0% (range: 21.4% to 35.7%). The study provides an important resource of information for healthcare professionals, researchers and policymakers working at the intersection of child psychiatry, pharmacogenomics and precision medicine.

## 1. Introduction

Although neurogenetic conditions are rare, their impact is substantial when viewed collectively. One approach to minimise disease burden could be using pharmacogenetics (PGx) to individualise treatment using established medication regimens. This approach could help identify genetic variants that affect drug metabolism (pharmacogenes) and reduce the frequency of adverse drug reactions (ADRs). However, most studies examining the clinical utility of PGx testing have been performed in the adult population [1]. Pharmacogenetic-guided prescribing can vary between children and adults due to developmental changes in metaboliser status and gene expression impacting drug responses [2]. Even when medication is initiated at small doses to potentially mitigate ADRs [3], this strategy may lead to poor clinical responses in the paediatric population [4].

The ontogeny of different drug-metabolising enzymes in the paediatric population has been described [5]. However, not all of these follow developmental profiles. For example, *thiopurine methyltransferase (TPMT), cytochrome P450 family 3 subfamily A member 5* (*CYP3A5)* and *sulfotransferase family 1A member 1 (SULT1A1)* have consistent expression patterns; hence, inferences from adult studies might be relevant [2,6]. In children, there is scope for genotype-guided prescribing [7], and the Clinical Pharmacogenetics Implementation Consortium (CPIC) guidelines have helped in advancing the clinical implementation of PGx [8]. In some disciplines, such as psychiatry, dosing guidelines have been made specifically for paediatric PGx prescribing [9], and in a study of 452 young people aged 11.9 ± 4.3 years, mood disorders and gastritis emerged as the main diagnostic groups for pre-emptive PGx testing [10]. Nonetheless, implementation of PGx dosing models in the paediatric population is still challenging [11]. Caution is needed when extrapolating data from studies carried out in adults. For example, while a systematic review and meta-analysis on the efficacy of PGx tests for *CYP2D6* and *CYP2C19* in depressive disorders in adults showed that PGx for treating depression was more useful than treatment as usual (TAU) [12] and improved remission of depressive symptoms [13], it is not known whether these findings would also be mirrored in the paediatric population. Broader cytochrome P450 (CYP) phenotyping in children in real-world settings is also suggested to vary, especially for those on multiple medications [14]. Associations in impulsivity-related networks within the brain, such as impairments in fronto-striatal functional circuits, suggest a link between antipsychotic induced weight gain in young adults (median age: 23 years) treated for psychosis [15]. However, there is no evidence whether these impulsivity-related brain networks would also be disrupted in the paediatric population.

Evidence indicates that in children and adolescents, Risperidone and Paliperidone tolerability is associated with CYP2D6 metaboliser status [16]. At the same time, pharmacokinetic modelling studies can help to identify associations between CYP2C19 metaboliser status and dosing with Escitalopram or Sertraline [17,18]. In individuals with Autism Spectrum Disorder (ASD), CYP2C19 ultrarapid metabolisers were associated with decreased tolerance [19], while advances in population pharmacokinetics of Atomoxetine in children and adolescents with attention deficit hyperactivity disorder (ADHD) may help to guide individualised prescribing [20]. A systematic review of 28 PGx studies in Autism Spectrum Disorder (ASD), of which 25 focused on young people, revealed limitations in PGx data in those with ASD [21]. While we have previously surmised that individuals with neurodevelopmental conditions are at higher risk of ADRs due to the severity of the condition, polypharmacy and comorbidity [22], there is uncertainty regarding the use of PGx in children. Indeed, in a survey of 282 paediatricians from America and Japan, less than 10% were familiar with PGx and those in America, <10% knew of CPIC [23]. Another survey of 958 child and adolescent psychiatrists in America showed the most frequently ordered genetic test was PGx (32.2%), and 73.4% rated PGx as “is at least slightly useful” [24]. However, 45% rated their knowledge of testing guidelines as “poor/very poor” [24]. This sentiment is also echoed by others, particularly when it comes to PGx testing following the use of psychotropic medications and long-term mental health outcomes in children [4]. Barriers for paediatric PGx implementation in the UK have been noted [1] and are a broader issue within the UK [25] but also across the PGx landscape globally [26]. Implementation challenges may particularly affect some diagnostic areas, such as clinical psychiatry [27]. Although CPIC guidelines provide evidence-based recommendations for *CYP2D6*, *CYP2C19* and *CYP2B6* to assist in antidepressant prescribing [28], there are other factors to consider for disorders involving the brain, in particular, the link between gut dysbiosis and increased risk of a neurodevelopmental disorder [29]. The gut microbiome can modify CYP450 enzymes [30]. Alterations in the microbiome can lead to changes in the efficacy of antidepressants [31], Risperidone [32], and other evidence suggests that there may be a relationship between microbiota expression and treatment resistance in depression and anxiety [33]. When all the data are viewed together, enriching the evidence base for further examining the genetic influences of psychotropic medication response in children would be useful. While some literature has explored PGx studies in ASD [21,34], there are limitations because not all medications used in ASD have PGx-based prescribing guidelines [35].

To our knowledge, no study has critically appraised the genetic influences on psychotropic medication response in children. As previously indicated, there is a fundamental need for separate PGx evidence for the paediatric population [1]. The focus of this review was therefore to systematically review the literature on PGx studies in children with a particular focus on psychotropic medication with a view to (I) better understand PGx associations in children and response to psychotropic medication, and (II) whether the emerging evidence could be helpful for future paediatric PGx implementation purposes. In doing so, the aim was to address the gap in the literature, which may also support the implementation of PGx-guided testing in children, particularly in mental health disorders.

## 2. Methods

### 2.1. Search Strategy

The search strategy followed the PRISMA guidelines for systematic reviews [36] and was used to search PubMed, Scopus, Cochrane, PsycINFO, Embase and Web of Science databases in February 2025. Two authors (A.M. and J.S.) searched independently in a blinded manner as previously described [37]. Searches included the truncation symbol to make the search terms as expansive as possible, and where appropriate, snowballing searching of references [37] was also used to identify as much of the pertinent literature as possible. For J.S., searching was performed from 2015 to February 2025, while for A.M., no start date filter was applied. The searches performed by A.M. were exported to Rayyan [38], where A.M. screened the studies by title and abstract, excluding all irrelevant literature. A scoping review of the literature using PubMed was also performed by J.S. in July 2025 using the terms “paediatric” or “youth” or “adolescent” to see if any further articles could be traced. After J.S. and A.M. completed the search and discussed eligible articles with G.W., a consensus was reached, and a final list of eligible articles was produced.

### 2.2. Search Terms

(pharmacogenomic* OR pharmacogenetic*) AND (brain*) AND (child*)

### 2.3. Population Characteristics

The database searches included all studies performed in the paediatric population.

### 2.4. Intervention

Studies that included information on PGx.

### 2.5. Eligibility Criteria

Inclusion Criteria
➢Records in full-text peer-reviewed journal articles.➢All information is for the paediatric population.

Exclusion Criteria
➢Records are not available in the English language.➢Studies performed in pre-clinical animal models.➢The following literature was excluded: reviews (all types), meta-analyses, preprints, letters, conference proceedings, clinical trial protocols and books.

### 2.6. Extraction of Data

Data from each eligible article were extracted into a table by J.S. and reviewed by A.M. and G.W. Information on country, study design, sample characteristics/demographics, assessment methods and relevant findings was extracted.

### 2.7. Thematic Analysis

This procedure employed a grounded theory approach to identify themes [39], utilising colour coding to manually identify text that represented patterns or similarities. This procedure was performed manually by J.S. using colour codes to identify words and text from the extracted data table. The coding framework has been previously described by the first author [37] and is a recognised method for identifying themes directly from data [39]. The extracted data from each eligible article was examined, and a colour was assigned if there were patterns or interconnections between different studies. Following this, G.W. independently completed a thematic analysis following the principles described in Braun and Clarke (2006) [40]. No formal statistical assessment of inter-coder reliability was completed; however, the second author, A.M., independently reviewed the completed thematic analyses from both J.S. and G.W. to assess the levels of agreement. Finally, all authors reviewed the themes and sub-themes and reached a consensus on the final themes that emerged. This process increased the methodological rigour of the thematic analysis. The frequencies of the final themes were displayed using Excel (Microsoft, 2023), and themes were presented in a table.

### 2.8. Quality Appraisal

The quality appraisal of the eligible articles was performed based on the Downs and Black checklist for reporting studies of health care interventions [41]. This checklist had recently been used for assessing the quality of articles in a systematic review of PGx testing to guide antipsychotic treatment in individuals aged 14 to 49 years [42]. The checklist has 26 items comprising of 10 items (items 1–10) for reporting, three items (11–13) for external validity, six items (items 14–19) for internal validity—bias, six items (items 20–25) for internal validity—confounding (selection bias) and one item (item 26) for statistical power. For this study, a modified version of the checklist described in Supplementary Table 2 in the work of Saadullah Khani et al. (2024) [42] was used. Items were scored as either Yes (1) or No (0) if the criterion was or was not met, respectively. However, item 5 “*Are the distributions of principal confounders in each group of subjects clearly described*?” was scored as Yes (2), Partially (1) or No (0). Each study could have a score ranging from 0 to 27 points. In some cases where the item was not relevant for the study, it was marked as not applicable (N/A). The quality appraisal was initially completed by J.S. and then independently performed by A.M. in a blinded manner. Both authors discussed the quality appraisal and reached an agreement before it was finalised.

### 2.9. Health Economic Evaluation

The health economic evaluation was based on the 24-item Consolidated Health Economic Evaluation Reporting Standards (CHEERS) checklist [43]. In this study, the health economic evaluation of all eligible articles was performed using the updated CHEERS checklist of 28 items [43]. The 28 items were organised into seven sections: title (item 1), abstract (item 2), introduction (item 3), methods (items 4–21), results (items 22–25), discussion (item 26) and other relevant information (items 27 and 28). Items were scored as either having one point if the criterion was met, no points if the criterion was not/partially met, or N/A if the item was not relevant to the study. The overall score was given as a percentage. We decided to perform the CHEERS checklist on the 50 articles. Initially, J.S. completed the CHEERS checklist, followed by A.M., who completed it independently. Both J.S. and A.M. reached an agreement before the health economic evaluation was finalised.

## 3. Results

Searching the six databases yielded 769 records: PubMed (*n* = 250), Scopus (*n* = 138), Cochrane (*n* = 4), PsycINFO (*n* = 10), Embase (*n* = 160) and Web of Science (*n* = 207). The PRISMA is shown in Figure 1. Following the identification process, 299 duplicates were removed, the titles and abstracts of 470 records were screened and 12 records were excluded. In total, 458 records were assessed for eligibility, and 31 full-text articles remained. A further 10 full-text articles were included after secondary “snowball” searching, and another 9 were identified by the second reviewer, A.M., and included after consensus agreement. In total, 50 full-text articles were included for analysis. Information from the 50 articles was synthesised into a Table (Table 1), and a thematic analysis was performed to reveal emerging themes.

### 3.1. Article Characteristics

Aspects related to different mental health conditions contributed to a significant proportion (30/50) of the studies described [22,35,44,46,48,50,51,52,54,57,58,59,61,62,63,64,65,66,67,68,69,70,73,74,75,78,79,80,83,84]. About half (24/50) of the studies [7,35,44,54,57,58,59,61,62,68,69,72,74,75,77,78,80,81,82,83,84,85,86,87] were performed in North America (United States and Canada), and the sample size ranged between 2 [44] and 2.9 million paediatric patients [7]. Most of the studies reported on ethnicity; however, some did not provide a breakdown of the ethnic groups within the study population [45,46,47,48,49,50,51,52,53]. Other studies only included Europeans [54] or European Caucasians [55,56] in their sample. Study designs ranged from multi-centre clinical trials [57,58,59] to retrospective case-controlled studies [47,49,52,60]. While most of the studies used genetic methods for assessment of genetic variants, some studies also used a review by a multidisciplinary team to assess the veracity of PGx testing in individuals with complex neurodevelopmental and neurobehavioural disorders [61] or neuroimaging as an adjunct to genotyping [62,63,64,65].

### 3.2. Thematic Analysis of the Analysed Studies

Thematic analysis of the 50 articles revealed five core themes (Table 2). The theme with the highest frequency was ‘*implications of non-CYP450 polymorphisms*’, which emerged from 22 articles, followed by ‘*paediatric CYP450 PGx*’, which emerged from 11 articles. Upon further analysis, three other themes were identified: ‘*Genetic predictors of response*’ (from eight articles) and ‘*insights for implementation and future research’* (from seven articles). The least common theme was ‘*phenoconversion’* that emerged from four articles. These themes are presented in Table 2, and each of the themes is described in the next section.


**
*Theme 1: Implications of non-CYP450 polymorphisms*
**


Theme 1 had the highest frequency, spanning 22 studies examining the role of non-*CYP450* polymorphisms. Upon analysis, this theme was further separated into two sub-themes, with “disorder-specific associations” emerging as the sub-theme with the highest frequency.


*
Sub-theme: Disorder-Specific Associations
*


Most of these studies were associated with neurodevelopmental and mental health disorders. Three studies also captured information related to oncology and were included due to the association with central nervous system (CNS) related adverse events. In neurodevelopmental disorders, genetic influences and their impact on onset, risk and treatment responses emerged as a prominent area of research.
*I, Neurodevelopmental Disorders**Epilepsy*

Epilepsy onset, susceptibility and response to treatment can be associated with genetic variants. Pharmacogenetic examination of genetic variants and susceptibility to epilepsy was evaluated in Egyptian children [60]. This study showed that children with the T-allele of ATP-binding cassette subfamily C member 2 polymorphism (ABCC2**rs717620*) had a higher risk of epilepsy when compared to healthy controls. Genetic variations were also observed between drug-resistant and drug-responsive individuals. Findings revealed no association between the 5,10-methylenetetrahydrofolate reductase (MTHFR) *rs1801133* polymorphism and response to seizure medications [47] or on the frequency of prescribed seizure medications [22]. Other studies also assessed seizure medication responsiveness [49,56]. Treatment resistance to Levetiracetam was shown to be associated with variants in the *synaptic vesicle glycoprotein* (*SV2*) gene [49]. Genotyping and allele distributions of the multidrug resistance protein 1 (*MDR1*) gene (*rs1045642* polymorphism) in Polish children demonstrated that the *MDR1 rs1045642* polymorphism is associated with treatment responses for epilepsy in children. Specifically, the T-allele may confer a protective role, while the C-allele was suggested to increase the epilepsy risk [56]. The *MDR1* gene, otherwise known as the *ATP-binding cassette subfamily B member 1* (*ABCB1*) gene, has also been shown to modify responses to psychotropic medications in children. In a PGx case comparison study, polymorphisms in the *ABCB1 gene* (G/G *rs1045642* genotype) had normal P-glycoprotein function. This makes the blood–brain barrier (BBB) less permeable to psychotropic medication. In comparison, the *ABCB1* gene (*A/A rs1045642* genotype) resulted in sub-optimal P-glycoprotein and a more permeable BBB [44].


*ASD*


One study used array comparative genomic hybridisation to assess novel variants in children with ASD. This study showed that copy number variation (CNV) and duplication of the T-box protein 1 (*TBX1*) gene might be implicated in ASD [46]. A further study showed that the G allele of the *rs4343* ACE polymorphism increased the risk of ASD [52].
*Attention Deficit Hyperactivity Disorder*

Two studies identified polymorphisms that alter Methylphenidate treatment responses in children and adolescents with attention-deficit hyperactivity disorder (ADHD). In combination with functional near-infrared spectroscopy (fNIRS), one study demonstrated that polymorphisms in the *Synaptosomal-Associated Protein 25* (*SNAP-25*) gene were associated with treatment efficiency of Methylphenidate in children with ADHD [66]. Another article examined the role of *latrophilin 3* (*LPHN3*) gene polymorphisms, proposing that *LPHN3* polymorphisms are associated with ADHD susceptibility and may also modulate Methylphenidate treatment responses [67].
*II, Mental Health Disorders**Anxiety and/or Major Depressive Disorder*

In a hypothesis-driven genetic study, interleukin (IL) polymorphisms were assessed to examine the association between different *IL* polymorphisms and aggression/internalising behaviours (depression or anxiety) in young children and adolescents [54]. The study identified the *IL6 rs2069827* polymorphism as having a statistically significant association with symptoms of depression. There were also trends observed for genetic variations in *IL1β* and *IL2* and their interactions with child adversity and depression. The tolerability of Sertraline, its titration and the maximum Sertraline dose utilised were shown to be associated with *HTR2A*, Solute carrier family 6 member 4 (*SLC6A4*) and Glutamate ionotropic receptor kainite type subunit 4 (*GRIK4*) polymorphisms in youth with anxiety and/or major depressive disorder [68].


*Bipolar Disorder*


Polymorphisms in *superoxide dismutase-2* (*SOD2*) were suggested to modulate white matter architecture in individuals with Bipolar Disorder (BD) [62]. This study was based on the premise of an altered redox imbalance in youth with BD. The G-allele of the *SOD2 rs4880* polymorphism had a higher antioxidant capacity compared to the A-allele. The G-allele was therefore suggested to be neuroprotective, and those individuals missing this allele were suggested to have a higher predisposition to oxidative stress [62].


*Obsessive Compulsive Disorder*


Genetic variation in polymorphisms associated with neurotransmitter pathways, i.e., glutamatergic and dopaminergic pathways, was shown to be associated with changes in white matter in children with OCD, providing further insights into the neurobiology of this disorder [65]. Another study used microsatellite markers [69] to explore candidate polymorphisms in young people with OCD. This study showed a role for Myelin Oligodendrocyte Glycoprotein (MOG) microsatellites and an increase in white matter volume in children with OCD.


*Acute Psychosis*


Retrospective genotyping data in 36 adolescents (mean age: 14.83 years) with an acute psychotic episode revealed that dopamine D2 receptor (*DRD2*) *rs1800497* and ATP Binding Cassette Subfamily B Member 1 (*ABCB1*) *3435C > T* polymorphisms modulate treatment responses and tolerability to antipsychotics [70]. Carriers of the *DRD2 rs1800497* polymorphism had a higher frequency of antipsychotic-related neurological symptoms.
*III, Oncology**Acute Lymphoblastic Leukaemia*

Genetic variation in treatment responses in children with acute lymphoblastic leukaemia (ALL) was assessed in two studies [53,71]. One study demonstrated that polymorphisms in the *ADORA1 adenosine A2A receptor* and *ADORA2A* gene were associated with methotrexate-related leukoencephalopathy in children with ALL. Those with the CC-allele of *ADORA2A rs2298383* polymorphism were shown to have a higher risk of methotrexate-related leukoencephalopathy [53]. Others demonstrated that polymorphisms are associated with chemotherapy-related CNS adverse events. In this study, *ABCB1 rs1128503 CC* or *rs2032582 GG* polymorphisms were shown to have a higher frequency of central nervous system (CNS) relapse in ALL [71].


*Brain Tumours*


Genotyping of 22 SNPs from nine genes associated with Methotrexate (MTX) pharmacokinetics demonstrated only a modest impact of SNPs on MTX metabolism in children with brain tumours [72].


*
Sub-theme: Treatment Response and Efficacy
*


Pharmacogenetic variation was shown to influence Risperidone responses, highlighting that specific polymorphisms are associated with altered drug exposure and side effects in children with ASD [50,51,73]. One study showed that the Brain-Derived Neurotrophic Factor (BDNF) *196G > A rs6265* polymorphism showed a statistically significant association with insulin resistance [50]. An earlier study from this group revealed that the Dopamine receptor D2 (*DRD2) Taq1A A2A2* polymorphism was important in hyperprolactinemia following Risperidone treatment [51]. Others demonstrated that the *rs78998153* polymorphism in the UDP-glucuronosyltransferase gene (*UGT2B17*) was the most predominant genetic marker for Risperidone exposure [73].


*Theme 1—Broader Literature Context*


Theme 1 highlights the role of genetic variation of non-*CYP450* polymorphisms across neurodevelopmental and mental health disorders in the paediatric population. It raises attention to non-*CYP450* polymorphisms and their influence on treatment responses, disorder susceptibility and adverse events. This is important because this area is often overshadowed by CYP450-focused studies. Studies investigating *ABCB1* genetic variants and their impact on psychotropic medication responses can provide further insights into drug mechanisms within the brain. Polymorphisms in this gene can affect the function of p-glycoprotein, which is found in the liver but also in the BBB, thereby affecting the permeability of drugs entering the brain [91]. The evidence synthesis showed that *ABC* variants can result in increased medication trials and hospitalisation [44], increased epilepsy risk [56,60] and modulate treatment responses to antipsychotics [70]. The broader clinical relevance of these findings is uncertain. When viewed collectively, studies examining the impact of *ABCB1* variants on psychotropic treatment responses and disorder susceptibility in children should be viewed as associative and not confirmatory. While some have indicated that specific *ABCB1* variants such as *rs2032583* have a role in modulating antidepressant responses [92,93], the ambiguity of what *ABCB1* variant to include in PGx gene panels limits their use in PGx-guided treatment strategies. No current PGx guidelines are available for recommending dosing adjustments based on *ABCB1* variants [91,94].

This theme also identified genes associated with neurotransmitter pathways such as serotonergic, dopaminergic and glutamatergic pathways [68,70], which have been linked more broadly to psychotropic efficacy. Under this theme, neurodevelopmental and redox genes were also shown to influence both psychiatric and neurodevelopmental risk. Variants in neurodevelopmental genes *TBX1,* BDNF, *SNAP-25* and *LPHN3* are implicated in neurodevelopmental risk and treatment responses [46,50,66,67]. Evidence suggests that SNAP-25 is associated with both neurological and neuropsychiatric disorders [95]. Variants in the redox gene (*SOD2*) showed an association with clinical variability in youth with BD [62] due to its modulation of white matter. In BD, a recent genome-wide study has identified 36 genes associated with BD, with GABAergic interneurons and medium spiny neurons implicated in the pathophysiology [96]. This finding underscores the notion that there are multiple genes involved in conferring both psychiatric and neurodevelopmental risk in the paediatric population.


**
*Theme 2: Paediatric CYP450 PGx*
**


This theme emerged from 11 articles and was split into three sub-themes, with the ‘treatment response and efficacy’ sub-theme emerging with the highest frequency.


*
Sub-theme: Treatment Response and Efficacy
*


In a retrospective PGx study in children with Tuberous Sclerosis Complex (TSC), the *CYP3A4* polymorphism (*CYP3A4*2*2) was shown to have an important role in the efficacy and manifestation of side effects following Everolimus treatment in these patients [45]. A further study from the PGx-SParK trial assessed *CYP2D6* genetic variation and amphetamine response in 214 children (mean age: 12 years) [59]. The authors showed that symptom improvement following Amphetamine treatment was dependent upon correction for phenoconversion of *CYP2D6*, and this could help to inform PGx-guided Amphetamine prescribing in young people. Altered CYP2D6 metaboliser status was also associated with an increased incidence of akathisia in youth with BD [74]. Young people with slow CYP2C19 metabolisers had poorer clinical outcomes (behavioural activation and weight gain) when treated with Escitalopram or Citalopram [75]. This study highlights the importance of CYP2C19 metaboliser status in treatment outcomes in young people given Escitalopram or Citalopram for the treatment of anxiety and/or depressive disorder. In neonates with patent ductus arteriosus (PDA), two *CYP2C9* polymorphisms (*rs2153628* and *rs1799853*) were identified to be associated with Indomethacin response and treatment of PDA [76]. This study suggested that genotyping *CYP2C9* polymorphisms could help manage Indomethacin toxicity and morbidity associated with PDA.


*
Sub-theme: CYP450 substrates and gene–drug pairs
*


This sub-theme emerged from three articles. A cross-sectional study of prescribing data in 16 healthcare systems from 2.9 million paediatric patients within the United States suggested that 1.3% (1300/100,000) of patients might require modifications to standard dosing regimens based on PGx prescribing guidelines alone [7]. This study also demonstrated that substrates for *CYP2D6* and *CYP2C19* pharmacogenes were the most frequently prescribed, accounting for 87.6% of actionable exposures based on current PGx recommendations [7]. In a study of prescribing data from 92 psychotropic medications in 787 individuals with ASD (mean age 15.4 years, 613 males and 174 females), 32% followed PGx-guided prescribing guidelines based on *CYP2D6* and/or *CYP2C19* pharmacogenes [35]. Charting dispensing rates for 57 drugs with PGx prescribing guidelines in Canada showed that gene–drug pairs with the most impact were associated with *CYP2C19* or *CYP2D6* variants [77].


*
Sub-theme: Disorder-Specific Associations
*


In a cohort of children with ASD with insulin resistance, *CYP2D6* genotypes do not influence plasma Risperidone concentrations [50]. Increased CYP2D6 metaboliser status (faster drug clearance) was shown to lower the odds of symptom improvement in young people treated with Fluoxetine for managing major depressive disorder or OCD [58]. Another study showed that the *CYP2C19* genotype was associated with Sertraline titration in youth aged <19 years with a diagnosis of anxiety or depression [68].


*Theme 2—Broader Literature Context*


Prescribing data from the United States show that *CYP2D6* and *CYP2C19*-associated drugs were responsible for 87.6% of PGx-guided prescribing in children [7]. In this group, antidepressants (Citalopram, Escitalopram and Amitriptyline) were amongst the drug classes that had the highest likelihood of having a clinically actionable impact [7]. Theme 2 aligned with the literature from the following perspectives. First, it showed that genetic variations in CYP450 genes can modify treatment responses in children with complex neurodevelopmental conditions such as ASD [19] and TSC [45]. Second, adjusting doses based on CYP450 metaboliser status can also improve the treatment responses in ADHD, anxiety and depression in young people [59]. Third, understanding the metaboliser status can help to minimise psychotropic-induced drug adverse events such as akathisia or behavioural activation [74,75] and provide further insights into Risperidone and Paliperidone tolerability [16]. When taken together, theme 2 further supports the premise that paediatric CYP450 PGx could be more thoroughly integrated into PGx guidelines, especially for the most high-impact gene–drug pairs (*CYP2D6* and *CYP2C19* variants).


**
*Theme 3: Genetic Predictors of Response*
**


This theme emerged from eight studies. In youth with anxiety and depressive disorders, the Sertraline dose is associated with the *HTR2A rs6313* polymorphism [68] while in OCD, the *MOG (TAAA)n* allele was shown to be associated with an increase in white matter volume [69]. Hypermethylation of non-coding ribonucleic acid (RNA) genes was also suggested to be a predictor of cognitive behaviour therapy (CBT) response in children with OCD [78].

In youth with BD, a study indicated that those with *5-hydroxytryptamine receptor 2A* (*HTR2A*) *A/A* or *A/G* genotypes might be associated with a higher risk of self-harm or harm to others [74]. Another study showed that polymorphisms in transcription factor binding sites (TFBs), such as the *TPH2 rs34517220* polymorphism, were significantly associated with improved symptoms of depression in children (mean age: 14.7 years) following Fluoxetine treatment [79]. The authors suggested that the *rs34517220* polymorphism, in particular, minor allele carriers could have a critical role in Fluoxetine responses.

Genetic variations in the *ataxia telangiectasia mutated* (*ATM*) and *organic cation transporter* (*OCT*) genes could be useful in predicting Metformin responses in children with ASD on antipsychotics [80]. One study demonstrated the role of *mu1 opioid receptor 1* (*OPRM1*) gene variants in morphine-induced respiratory depression (MIRD) in young people following surgery for spine fusion [81]. This study showed that those with the *OPRM1 A118G AA* genotype have a higher risk of MIRD (*p* = 0.030). Genetic variants in the *ABCB1* gene were also shown to influence intravenous morphine responses for pain management in children undergoing tonsillectomy. In this study, children with the *ABCB1 rs9282564* GG and GA genotypes had an increased risk of respiratory depression and prolonged hospitalisation [82]. Interestingly, having an additional copy of the G allele increases the odds of hospitalisation due to post-operative respiratory depression by 4.7-fold [82].


*Theme 3—Broader Literature Context*


A balanced approach, acknowledging epigenetic markers, is needed when placing this theme into context with the broader literature. Evidence from the theme indicated that methylation profiles may help in predicting treatment responses. Hypermethylation of non-coding RNA can modify CBT responses in children with OCD [78], and this aligns with the broader literature for the role of dysregulated methylation in psychiatric disorders [97]. Both genetic and environmental factors are known to imbalance the epigenome in ASD [98]. When viewed together with the literature, this theme underscores a role for epigenetics as an essential factor when considering genetic predictors of treatment response in the paediatric population. Genetic variants in *ATM* and *OCT* highlight the impact of other genes outside the sphere of mental health for managing antipsychotic weight gain [80]. However, the clinical relevance of this study should be tempered because the small sample size of participants on Metformin (n = 26) limits the broader generalisation of the findings. At present, there is insufficient evidence to demonstrate a consistent role for ATM and OCT in antipsychotic weight gain.


**
*Theme 4: Insights for Implementation and Future Research*
**


Implementation of PGX is a significant challenge, and the “insights for implementation” theme emerged from seven studies. Findings suggest a multidisciplinary model could help guide paediatric PGx clinical decision-making in cases involving complex neurobehavioral and neurodevelopmental profiles [61]. Neurobiology and neuroimaging markers could also be useful adjuncts alongside PGx testing for predicting the clinical trajectory of psychosis in individuals [63,64]. One study in youth aged 13–18 years with a major depressive disorder examined the clinical impact of PGx. This study showed no difference between the PGx and TAU arms [83]. It led to the premise that a PGx strategy focusing on reducing medication side effects rather than on improving medication efficacy could be an alternative option to consider [84]. In a survey of 1358 healthcare providers comprising 37 different specialities, paediatric specialities rating PGx testing as most clinically useful were haematology/oncology (63.8%), anaesthesiology/pain and surgery (59.5%), followed by psychiatry (46.7%) [85]. This study also showed that unfamiliarity with PGx testing (78.2%), followed by cost/insurance coverage (62.8%), application to clinical practice (62.5%) and doubts about the clinical value of PGx testing (60.2%) were the most identified challenges [85]. Finally, others have indicated that pre-emptive testing could help those from high-risk services, potentially at higher risk of ADRs with standard dosing regimens [86].


*Theme 4—Broader Literature Context*


Interestingly, the theme pointed towards using PGx to improve the safety and tolerability of medications instead of increasing efficacy [83,84]. A study of 213 adult outpatients (n = 105 received PGx-guided dosing and n = 108 received TAU) in a community pharmacy setting demonstrated that PGx-guided prescribing for antidepressant treatment improves outcomes for depression, anxiety and disability when compared to TAU [99]. Results from the Genomics Used to Improve DEpression Decisions (GUIDED) trial demonstrated that although PGx testing improved response and remission rates when compared to the usual standard of care in individuals >18 years of age with depression, it did not reach statistical significance in symptom improvement [100]. However, a sub-analysis of the GUIDED trial focusing on those taking medications predicted to have gene-drug interactions demonstrated improvements in response, remission rate and symptom improvement compared to TAU [101]. It is too premature to suggest whether PGx could be used as a tool for reducing adverse events instead of improving treatment responses or remission rates in the paediatric population. Such a reframing would require more evidence in the paediatric population before conclusions can be made. The notion for improved clinical education and health system readiness when using PGx to improve treatment outcomes was also highlighted in this theme. A multidisciplinary model to facilitate PGx-guided decision making would be beneficial for managing complex cases in real-world settings to minimise multiple medication trials [61]. This roadmap has also been adopted in other settings [102,103].


**
*Theme 5: Phenoconversion*
**


Phenoconversion emerged from four studies. Two were Canadian studies from the Pharmacogenetic-Supported Prescribing in Kids (PGx-SParK) trial. One of these consisted of 1281 young people (mean age: 13.5 years) and showed that 46% of the cohort receiving medication for mental health conditions had phenoconversion for one of four pharmacogenes: *CYP2B6*, *CYP2C19*, *CYP2D6* or *CYP3A4* [57]. Furthermore, 24% had at least one clinically actionable phenoconversion that changed standard prescribing, i.e., adjusting the dose or switching medication. Phenoconversion mainly was related to substrates for *CYP2D6* (17%), *CYP2C19* (9%) and *CYP2B6* (4%). Substrates for *CYP3A4* (<1%) had minimal impact [57]. Another study showed that after correcting for phenoconversion, CYP2D6 poor metabolisers had a 3.67x greater odds of reporting symptom improvement with Amphetamine when compared to intermediate metabolisers [59]. However, after adjusting for study confounders, phenoconversion did not impact self-reported side effects [59], which agrees with others [58]. Others have also observed phenoconversion of *CYP2D6*. In 27 children with neurodevelopment/neurobehavioural disorders, 25% had phenoconverted to a CYP2D6 poor metaboliser, and 66.7% had clinically actionable PGx recommendations [61].


*Theme 5—Broader Literature Context*


Evidence from the literature indicates that phenoconversion poses a significant challenge in the interpretation of paediatric PGx. The theme indicated that phenoconversion is common in mental health disorders, mainly with drugs associated with CYP2D6. Failing to adjust for phenoconversion can cause changes in drug exposure, leading to poor treatment responses or increased side effects. For CYP2D6, more than 170 haplotypes (or star [*] alleles) have been identified [104], which can significantly affect CYP2D6 drug metabolism [105]. In a systematic review of studies that examined cytochrome P450 metabolism, factors that influenced phenoconversion alongside CYP450 inducers/CYP450-inhibiting drugs were age, inflammation and vitamin D exposure [106]. Recognising phenoconversion in the paediatric population is crucial for improving therapeutic outcomes, especially those with multiple neuropsychiatric/neurodevelopment disorders and treatment resistance. Tools may assist in the assessment of phenoconversion with CYP2D6 [107].

### 3.3. Quality Appraisal and Health Economic Evaluation


*
Quality Appraisal
*


A quality appraisal of the 50 studies is presented in Appendix A. The average score from the studies was 17.7, ranging from 14 points to a maximum of 24. Most of the studies performed well when critically appraised against the 10 reporting items, i.e., reporting objectives, patient characteristics, describing confounders and main findings and reporting on actual probability values. However, only 17% (8/48) of studies [49,54,55,62,66,67,72,83] had information describing patient characteristics that were lost to follow-up. Similarly, when scored against the six items for internal validity (bias), most of the studies scored poorly (either 0 or unable to determine [UTD]) when it came to blinding study subjects to the intervention or blinding those measuring the main outcomes. Lower scores were also revealed after critically appraising studies against the six items for internal validity (confounding—selection bias). Four (4) studies [72,80,83,84] had information on whether study subjects were randomised to intervention groups, with most of the studies scoring UTD for this item. Finally, 16% (8/49) of the studies [54,60,71,79,80,81,82,83] had a power calculation (Appendix A). In one of these studies [80], a power calculation could not be identified; however, the secondary analysis that investigated three-way treatment interaction did reduce the statistical power by half.
*Health Economic Evaluation*

Using the 28-item health economic evaluation CHEERS checklist, the average score was 33.0%, ranging from 21.4% to 35.7% (Appendix A). None of the 50 studies analysed had a health economic plan. Patient and public involvement to inform the study design was also not reported in any studies. Other items on the checklist fared better because most studies were able to summarise the key findings and describe their limitations. Seven (7) studies [35,48,51,60,63,64,70] failed to report on funding requirements, and three [50,63,79] did not report their conflict of interest (Appendix A).

## 4. Discussion

This study is a comprehensive analysis of genetic influences on psychotropic medication responses in children. The evidence synthesis provides an enriched data set analysing 50 studies containing between 2 and 2.9 million individuals. The study showed that 60% of these articles analysed data on young people with a specific mental health condition. Thematic analysis revealed five themes, with the most dominant theme related to *non-CYP450* polymorphisms in paediatric disorders emerging from 22 studies, followed by a theme for paediatric cytochrome P450 PGx. Genetic predictors of response and insights for implementing PGx and future research were other important themes that emerged. In the paediatric setting, PGx was the most clinically useful in haematology/oncology, anaesthesiology/pain and surgery specialities. Just under half of raters (46.7%) stated that PGx would be clinically useful in psychiatry. Barriers to implementing PGx in the paediatric population include cost, clinical application and value of testing. Drugs that are metabolised via CYP2D6 and CYP2C19 accounted for most actionable exposures in the paediatric population. In youth with mental health conditions, nearly half had phenoconversion, which was primarily associated with substrates for CYP2D6, followed by CYP2C19.

Despite these findings that are more influenced by *CYP2D6* and *CYP2C19* pharmacogenes, our thematic analysis showed that assessing non-*CYP450* polymorphisms, especially in those with mental health conditions, would also be helpful. The study data also showed that epilepsy onset, risk and response to treatment were associated with non-*CYP450* genetic variants [44,49,56,60,87]. This was also seen in oncology studies that examined different polymorphisms [53,71,72,88]. Some studies also bridged across both CYP450 and non-*CYP450* polymorphisms, such as examining the role of *CYP2C9*, adrenoceptor (*AR*) or Gs protein α-subunit gene (*GNAS*) polymorphisms in indomethacin [76] or dobutamine [89] responses in neonates. This multi-gene approach could be adopted more broadly and underscores the notion that focusing mainly on metabolic enzymes (CYP450) precludes meaningful insights from other non-*CYP450* genetic variants from being identified. Such an approach may be especially pertinent for complex psychiatric and neurodevelopmental disorders such as ASD and BD, where non-*CYP450* genetic variants affecting the BBB, neurotransmitter and redox pathways could have an integral role.

### 4.1. Developmental Considerations

Paediatric PGx differs from adult populations due to the ontogeny of drug-metabolising enzymes and developmental neurobiology. Several factors can influence the ontogeny of drug-metabolising enzymes. Activity of CYP2C19 increases during early infancy, and 50–75% of adult levels are not reached until 5 months, where levels peak [108,109]. In contrast, the activity of CYP2D6 peaks after birth and remains stable until adolescence [108]. Moreover, in neurotypical children, age and puberty have little impact on CYP2D6 enzyme activity, and CYP2D6 genotype remains the most prominent factor of variability in CYP2D6 activity during adolescence [110]. Several factors can influence the developmental trajectory of CYP450 enzyme pathways, and these have been described in detail elsewhere [111]. Nevertheless, evidence is limited when considering the developmental trajectory of drug-metabolising enzyme pathways in children with mental health conditions such as ASD and ADHD who are exposed to multiple medications. Phenoconversion of CYP450 enzymes can be affected by co-administered drugs, disease states or environmental factors [112]. In youth with mental health disorders, substrates with an actionable CYP2C19 and CYP2D6 phenoconversion increase with age, and the strength of the relationship was greater for CYP2C19 [57]. Evidence indicates that CYP450 enzymes are downregulated in some inflammatory conditions [112], which could have relevance for neurodevelopmental disorders [113]. Other data suggest that interactions between polymorphisms in interleukin genes and childhood adversity can change the symptoms of depression [54]. When viewed together, these studies highlight how genetic variants of inflammatory biomarkers can modulate early life experiences that affect mental health outcomes. In summary, drug-metabolising enzyme pathways and developmental neurobiology need to be viewed alongside disease states and phenoconversion when applying paediatric PGx.

### 4.2. Heterogeneity in Studies

The most prominent theme was related to non-CYP450 genes, and while it highlights that these genes can modulate the effects of psychotropic medications, the heterogeneous nature of the studies limits adding weight to the evidence in the individual themes. Other genetic factors such as CNV [46], microsatellite markers [69] and hypermethylation of non-coding RNA [78] were included, and the study designs were diverse, ranging from case reports to observational studies, randomised controlled trials and population-level studies. Although a variety of psychotropic medications were evaluated, some studies also evaluated chemotherapeutics [53,88] and analgesics [76,81,82]. A variety of outcome measures were also used, such as neuroimaging biomarkers [63,64,65] and hospitalisation rates [44,82]. Treatment duration and dosing schedules of psychotropics, use of multiple psychotropics in different studies and their impact on non-*CYP450* and *CYP450* genes can also contribute to methodological differences across studies. In addition, data from 24/50 of the articles were derived from studies completed in North America; therefore, the findings may not be generalisable to the wider population. Moreover, while the present evidence synthesis reported on study ethnicity, it showed that some studies did not report on the ethnic group of participants or only included specific groups. This limits the outreach of the findings, especially when applied to populations from diverse backgrounds, and draws attention to expanding paediatric PGx studies to individuals from underrepresented populations, as highlighted by others [114]. In summary, while the complexity of evidence is reflected by the heterogeneous nature of studies included in the themes, our broad coverage of paediatric PGx, capturing both *CYP450* and non-*CYP450* variants, enriches the overall evidence base and improves our understanding of genetic influences on psychotropic medication response in diverse paediatric populations.

### 4.3. Health Economics and Quality Appraisal

Various analysis tools to assess health economic evaluations are available [115]; however, to limit variability in assessment tools for health economic evaluation (and quality assessment) in general, we chose to use a method that had already been used to assess the economic evaluations of PGx testing for antipsychotic treatment [42]. The economic evaluation of studies revealed that none of the studies had cost-effectiveness included. Hence, further details of health economic models, such as resource use and costs and valuation of outcomes, could not be assessed. This finding highlights a significant gap in the literature, and in real-world terms, a CHEERS score of 33% demonstrates that the studies analysed in this comprehensive review have limited use in informing healthcare discussions regarding the cost-effectiveness of paediatric PGx. We are therefore unable to assess whether PGx-guided treatment, especially for psychotropic medication in the paediatric population, is better than TAU. In psychiatry, “*evidence for the cost-effectiveness of PGx testing*” was identified as a highly cited barrier in 46 studies examined [116]. A major factor influencing this was the test used in clinical practice [116]. Health economic evaluation in paediatric PGx testing also needs to account for the test used and the impact it would have on economic utility. A report by the Royal College of Psychiatrists (2023) has indicated that there is little clinical benefit for PGx testing of *CYP2D6, CYP2C19* or other genes for psychotropic medication prescribing [117]. Different factors can influence treatment outcomes. Indeed, combinatorial classification and regression tree (CART) analyses have indicated that treatment with Sertraline in paediatric anxiety and depressive disorders needs to consider a combination of PGx-guided dosing and clinical factors such as age, race, diagnosis and concomitant medications to optimise treatment outcomes [68].

The quality appraisal of the studies revealed an average score of 17.7/27 points, which indicates an overall “fair” rating for the methodological quality of the studies [118]. This rating shows that while most of the studies were robust in their reporting, the methodological rigour was varied across the studies and demonstrated limitations. Patient and public involvement was not discussed in any of the studies, and only eight studies had a power calculation. When viewed together, the health economic and quality appraisal results suggest that future studies need to address shortcomings to better determine the clinical utility and health economic value of PGx-guided testing in children, and an individualised approach to testing probably needs to be adopted [119].

## 5. Limitations

While our study provides an enriched synthesis of information, there are several limitations. First, our review was not registered. Second, while the search was focused on the child population, some of the included studies had older subjects [22,90]. Third, we acknowledge that the systematic review search terms may be limited and may not capture all relevant literature. While a broader search strategy could have traced other literature, we wanted to focus the search on the genetic influences of psychotropic medication in children. Fourth, the thematic analysis coding frameworks did not include a formal statistical assessment of inter-coder reliability. While inter-coder reliability is considered good practice, it is not universally accepted, with some suggesting that it may be an unnecessary step [120]. The thematic analyses were performed independently by J.S. and G.W. and then reviewed by A.M. Emerging themes were then discussed before a consensus was reached. This procedure increases the methodological rigour of the process. Notwithstanding these limitations, a thematic analysis of 50 articles did allow us to provide a comprehensive narrative on the genetic influences of psychotropic medication in children. To minimise searching bias, two authors (J.S. and A.M.) also independently searched the literature in a blinded manner. Furthermore, health economic and quality appraisal data were independently reviewed by (J.S., A.M. and G.W) before a consensus was reached. Finally, not all the studies had genetic variants associated with PGx recommendations. Therefore, we cannot determine whether these genetic variants are clinically meaningful and meet the criteria to be included in future regulations or guidelines. Nevertheless, we hope that the findings from our comprehensive analysis have enriched the evidence base and provide a valuable resource when examining the role of genetic associations and their impact upon psychotropic medication use in children.

## 6. Conclusions

This comprehensive analysis revealed five themes. Non-*CYP450* polymorphisms in paediatric disorders were the central theme and suggested that focusing on this group of genetic variants could facilitate the development of clinical pathways for genetic testing in this patient group. Unsurprisingly, the second most common theme that emerged was associated with cytochrome P450 PGx, as *CYP2D6* and *CYP2C19* pharmacogenes are metabolised by most psychotropic medications. The thematic analysis revealed that a multidisciplinary model could facilitate the implementation of PGx testing [61]. This approach is also welcomed by the Royal College of Psychiatrists, which has suggested that the development of clinical pathways for genetic testing should adopt a multidisciplinary approach involving Child and Adolescent Mental Health Services (CAMHS), community paediatrics and genomic medicine [117]. Pathways of PGx in child and adolescent psychiatry are summarised in Figure 2.

In adults with major depression, improvement based on PGx-guided treatment was more consistent in those with 1–3 failed medication trials [121], and this strategy could be adopted in the paediatric population to streamline pathways and specifically target individuals with failed medication trials. Focusing on reducing medication side effects rather than medication efficacy is another option to consider, as it could improve medication tolerability and reduce medication switching. This premise was assessed in a study of 170 adolescents (aged 13–18 years) with depression. It demonstrated the clinical utility of using PGx testing to prioritise side-effect burden rather than medication efficacy [84]. However, limitations in data in the paediatric population mean that the clinical usefulness of reframing PGx strategies that prioritise side-effect burden instead of medication efficacy remains uncertain. Education and training can help to improve uptake of PGx testing [122], and others have suggested strategies for PGx implementation into clinical practice [123]. In conclusion, our systematic review provides an essential information resource on the genetic influences of psychotropic medication responses in children.

## Figures and Tables

**Figure 1 genes-16-01055-f001:**
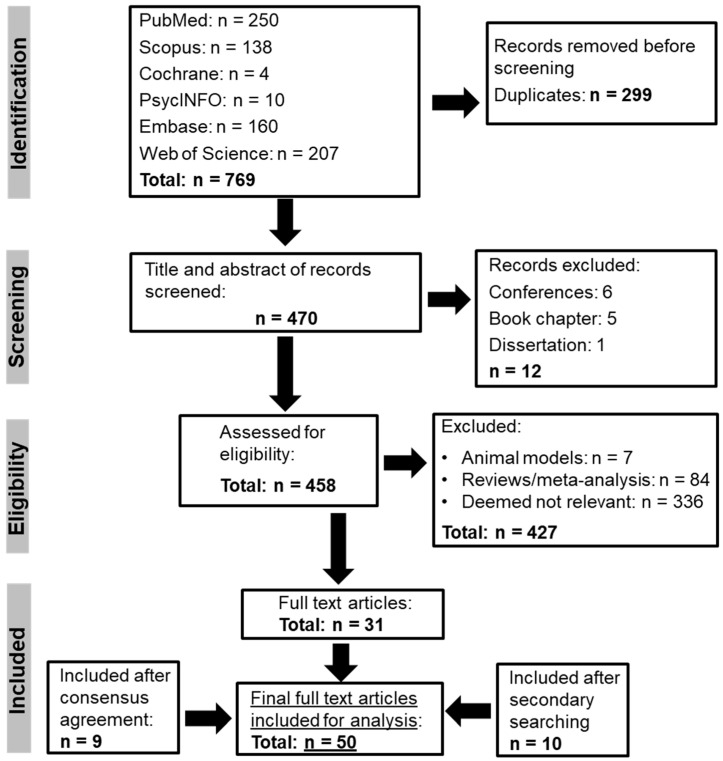
PRISMA flowchart.

**Figure 2 genes-16-01055-f002:**
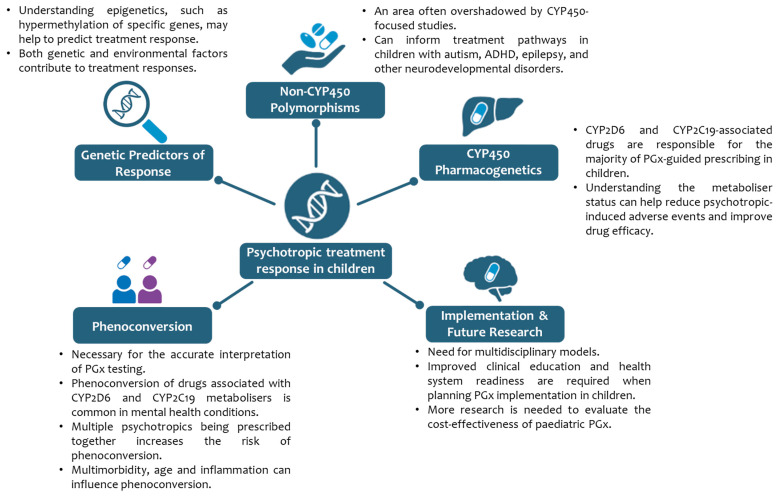
Pathways of pharmacogenetics in child and adolescent psychiatry. Abbreviations: ADHD: attention deficit hyperactivity disorder; CYP450: cytochrome P450; CYP2D6: cytochrome P450, subfamily 2D, polypeptide 6; CYP2C19: cytochrome P450, subfamily 2C, polypeptide 19; PGx: pharmacogenetics. This figure was created using images from BioRender (https://biorender.com/).

**Table 1 genes-16-01055-t001:** Summary of eligible studies.

Source	Region	Ethnicity Reported (Yes/No)	Study Design	Sample Characteristics	Assessment Methods	Relevant Findings
Ramsey et al. (2020) [7]	^a^ United States—IGNITE Pharmacogenetics Working Group	Yes	Cross-sectional study of prescribing data from 16 healthcare systems	The study group consisted of data from 16 health systems with approx. 2.9 million paediatric patients aged less than 21 years (range: 3 to 14 years). The prescribing frequencies were based on CPIC level A evidence (38 CPIC level A drugs and 20 associated genes) between 2011 and 2017.	The main outcome measure was the frequency of level A prescribing and actionability.	In the paediatric population, prescription of CPIC level A drugs is common (8000–11,000 per 100,000) About 1.3% of patients (1300/100,000) may require modification to standard treatment regimens based on PGx guidelines. Substrates for *CYP2D6* and *CYP2C19* were the most frequently prescribed and accounted for 87.6% of actionable exposures.
Singh et al. (2024) [22]	United Kingdom	Yes	Observational study	Sample size was *n* = 65 with a mean age (min; max) of 18.7 (2.5; 47.5) years Diagnoses consisted of 60 with RTT and 5 with atypical RTT	PGx testing for MTHFR *rs1801133* and *rs1801131* polymorphisms Assessment of clinical severity	The study showed that those with the homozygous MTHFR genotype were more impaired than those carrying the non-homozygous genotype. MTHFR *rs1801133* and *rs1801131* polymorphisms are associative genetic modifiers of clinical severity in RTT.
Ahmed et al. (2022) [35]	Canada	Yes	Observational study	Prescribing data were reviewed in 787 (*n* = 613 males) cases with ASD (mean age [range]: 15.4 years [2–87 years]) during 2012 and 2014	Retrieval of prescribing data from 92 psychotropic medications Prescribing data were referenced against published PGx guidelines	The study showed that 58% and 37% were prescribed at least one or two or more psychotropic medications, respectively. Methylphenidate (16.3%), followed by Risperidone (12.8%) and Lorazepam (12.1%), was the most frequently prescribed psychotropic medication. About 30% of the drugs prescribed were associated with PGx prescribing guidelines, the majority based on *CYP2D6* and/or *CYP2C19*.
Kalla et al. (2023) [44]	United States	Yes	Case comparison report to investigate *ABCB1* polymorphisms and blood–brain barrier (BBB) access to psychotropic medications.	Two paediatric patients aged 9 (case A) and 11 (case B) years old. Cases A and B were selected from a cohort comprising 130 paediatric patients who had PGx testing performed by psychiatric providers during 2017–2020.	Genotyping was performed on 14 pharmacogenes (*ABCB1, COMT*, *CYP2C9*, *CYP2C19*, *CYP2D6*, *DBH*, *DRD1*, *DRD2*, *DRD4*, *GABRG2*, *GAL*, *5HTR2A*, SLC6A4) Review of medical records	Following genotyping, the study showed that Case A had a functional *ABCB1 gene* (G/G *rs1045642* genotype), suggesting normal P-glycoprotein function. This genotype makes the BBB less permeable to psychotropic medication. Case B had a sub-optimal genotype of the *ABCB1* gene (A/A rs1045642 genotype), leading to a sub-optimal P-glycoprotein and a more permeable BBB. More medication trials and dose adjustments were noted for Case A than for Case B. Case A also had more hospitalisations than Case B. Overall, the study showed that PGx testing of the *ABCB1* gene can inform psychotropic medication prescribing and may help to better understand the clinical trajectory of patients.
Concha et al. (2023) [45]	Spain	No	Retrospective observational study	Ten (10) participants diagnosed with TSC aged 9.5 years (4.25–12) All were receiving Everolimus (oral) for management of TSC.	Genotyping of drug-metabolising genes for Everolimus Ten (10) genes were analysed.	The key finding from the study indicated that polymorphisms affecting *CYP3A4* are implicated in Everolimus pharmacokinetics. Valproic acid (CYP3A4 inhibitor) treatment and combined *CYP3A4* poor metaboliser phenotype status led to poorer outcomes. This finding suggests that the *CYP3A4* polymorphism may have a critical role in the efficacy and manifestation of side effects following Everolimus treatment.
Alhazmi et al. (2022) [46]	Saudi Arabia	Yes ^¥^	Observational study	In total sample size was 19 in children aged 3–12 years of age. Patient (ASD) sample (*n* = 15): Male (*n* = 12) and female (*n* = 3) Control sample (*n* = 4): Male (*n* = 3) and female (*n* = 1)	Samples of DNA were analysed using DNA sequencing and genomic hybridisation.	Copy number variations in chromosome 22 were found in six ASD samples. Genomic hybridisation revealed *TBX1* gene mutations in ASD samples but not controls.
Alyoubi et al. (2022) [47]	Saudi Arabia	Yes	Multicentre case-controlled retrospective study	Screening consisted of *n* = 200 participants with *n* = 150 approached and *n* = 49 did not finish. In total, 160 participants were recruited. ➢ Patient (epilepsy) sample (*n* = 101): age (SD) 7.3 ± 4.1, female (*n* = 33) and male (*n* = 68) ➢ Control sample (*n* = 59): Age (SD) 8.8 ± 3.2, female (*n* = 20) and male (*n* = 39)	Genotyping for MTHFR (*rs180133*) SNP	This study examined the association between the MTHFR *rs180133* SNP and seizure medication response. There was no statistically significant association between drug response and epilepsy risk in those paediatric patients with the MTHFR *rs180133* SNP.
Firouzabadi et al. (2022) [48]	Iran	No	Cross-sectional study	The sample consisted of 83 paediatric patients with ASD aged 3 to 12 years of age referred between 2013 and 2015. All children were treated with Risperidone and not on any psychotropic medications 6 months before participating in the study.	Genetic analysis was performed on two *ACE I/D* polymorphisms (*rs4343* and *rs4291*). Response to Risperidone was assessed at baseline, 4 and 12 weeks.	The study suggested that ACE gene polymorphisms were not associated with response to Risperidone treatment. The findings suggested that the renin–angiotensin system is not involved in Risperidone treatment response in children with ASD.
Wolking et al. (2020) [49]	^b^ European consortium	No	Case-controlled study	Sample comprised of 1622 individuals with a median onset of epilepsy at the age of 15 ± 15.6 years Study participants were exposed to common AEDs.	Review of electronic case report form Genotyping using WES	Assessment of whether genetic variants could have a role in resistance to ASMs, the study showed that genetic variants with missense and truncating mutations were found in those resistant to Valproic acid. Those with variants in the synaptic vesicle glycoprotein family gene were shown to be resistant to Levetiracetam.
Sukasem et al. (2018) [50]	Thailand	No	Observational study	The sample consisted of 89 participants (81 males and 8 females) with ASD taking daily doses of Risperidone The median age (IQR) was 10.0 (8.9–13.4) years.	Genotyping analysis of the following genes: *ABCB1*, *BDNF*, *CYP2D6*, *DDR2*, *GHRL* and *LEP* Measurements of plasma Risperidone	The study examined PGx of Risperidone-induced insulin resistance in children with ASD. About 17% of participants had insulin resistance. Among the genes assessed, the BDNF *196G>A rs6265* polymorphism was associated with insulin resistance (*p* = 0.025) and suggests that this polymorphism could be a genetic marker for predicting insulin resistance in children with ASD treated with Risperidone.
Sukasem et al. (2016) [51]	Thailand	No	Retrospective cross-sectional study	Sample comprised of 147 children aged 3–19 years with ASD treated with Risperidone. Subjects were recruited between May 2012 and April 2013 and were receiving Risperidone for at least 6 weeks.	Genotyping of *CYP2D6* and *DDR2* polymorphisms Measurement of serum prolactin	The study showed that there was no statistically significant correlation between *CYP2D6* genotypes were serum prolactin concentrations. Those with the *DRD2 Taq1A A2A2* polymorphism had statistically significant differences in prolactin concentrations. In summary, the data indicated that DRD2 Taq1A A2A2 polymorphisms have an important role in hyperprolactinemia following Risperidone treatment.
Firouzabadi et al. (2016) [52]	Iran	No	Case-controlled study	The study sample consisted of 120 participants with a mean (SD) age of 7.5 ± 2.8 years (86 males; 34 females). All participants had a clinical diagnosis of ASD.	Genotyping of two polymorphisms (*rs4291* and *rs4343*) within the *ACE* gene	The G allele of *rs4343* was shown to increase the risk of ASD x1.84 fold in comparison to those with the A allele (*p* = 0.006). Those with the D allele had an increased risk of ASD by x2.18 fold (*p* = 0.006). In summary, genotyping revealed that the genetic diversity of the renin–angiotensin system is associated with an increased risk of ASD in children.
Tsujimoto et al. (2016) [53]	Japan	No	Observational study	The study sample consisted of 63 Japanese participants aged between 10 months and 15 years of age. Seven (7) were excluded, and fifty-six patients’ data were analysed for the study. All patients were treated for ALL or lymphoma.	Genotyping analysis Review of medical data Assessment of methotrexate-related leukoencephalopathy	The study indicated that high-dose systemic methotrexate treatment in individuals with the CC genotype at *rs2298383* in *ADORA2A* was associated with a higher risk of methotrexate-related leukoencephalopathy. The authors suggest that targeting the adenosine pathway could help to mitigate the effects of methotrexate-related leukoencephalopathy in children with ALL.
Pouget et al. (2021) [54]	Canada	Yes	Hypothesis-driven genetic study	The clinical sample consisted of 255 cases (male: *n* = 152) and 226 controls (male: *n* = 125). Cases were aged 6–16 years and referred to an outpatient psychiatric service for major aggression.	Genotyping of *IL1B*, *IL2* and *IL6* gene variants Assessment measures for clinical characteristics (internalising problems, anxiety, depression and childhood adversity)	The study indicated that there was no association between *IL1B*, *IL2* and *IL6* gene variants and childhood-onset aggression. The *IL6 rs2069827* polymorphism was associated with depressive symptoms. A trend was noted for *IL1β* and *IL2* polymorphisms and depression with childhood adversity.
Kukec et al. (2021) [55]	Slovenia	Yes	Observational study	The study consisted of: Participants with epilepsy and/or cerebral palsy (*n* = 229), of whom 95 had perinatal HIE. Healthy controls (*n* = 129)	Clinical characteristics Genotyping of *HIF1A rs11549465* and *rs11549467* polymorphisms	The study showed that there was no statistically significant association between the HIF1A *rs11549465* and *rs11549467* polymorphisms and an increased risk of epilepsy, its drug resistance or cerebral palsy after neonatal HIE. Clinical features were found to be the best predictor of neurological outcome following HIE, and the authors suggest that HIF-1 could be implicated in the onset of epilepsy, but not at the level influenced by gene polymorphisms.
Stasiołek et al. (2016) [56]	Poland	Yes	Observational study	Drug-resistant epilepsy cohort (*n* = 106): ➢ Mean age: 8.5 ± 4.8 Drug responsive epilepsy cohort (*n* = 67): ➢ Mean age: 8.2 ± 4.0 Healthy control cohort (*n* = 98): ➢ Mean age: 8.3 ± 4.6	Genotyping and allele distributions of the *MDR1* gene (*rs1045642* polymorphism)	The C allele was shown to confer a higher risk of drug-resistant epilepsy, while the T allele was suggested to have a protective role. The study showed that the *MDR1 rs1045642* polymorphism is associated with responsiveness to epilepsy treatment in children.
Gerlach et al. (2025) [57]	Canada	Yes	^c^ PGx-SParK clinical trial	The study group consisted of 1281 young people (age range: 6 to 24 years). Mean age (SD): 13.5 years (3.9), 49.3% were female.	Genomic DNA was obtained from saliva samples All participants received psychotropic medication and had a PGx test for four pharmacogenes (*CYP2B6*, *CYP2C19*, *CYP2D6* and *CYP3A4*).	In young people receiving medication for mental health conditions, about half (46%) have phenoconversion. Of these, 24% had at least one clinically actionable phenoconversion leading to a change in standard prescribing, mostly related to CYP2D6 (17%) and CYP2C19 (9%) substrates. Phenoconversion of *CYP2C19* and *CYP2D6* accounted for a 7.3- and 4.5-fold increase in poor metabolisers, respectively. Phenoconversion for *CYP2C19* and *CYP2D6* was said to increase with age. In the sample, 39.1% were taking a CYP2C19 inhibitor while 24.4% were taking a CYP2D6 inhibitor. The most frequent inhibitors were fluoxetine (20.1%), cannabidiol (16.9%) and fluvoxamine (8.2%), while glucocorticoids were the most frequent inducers (2.1%).
Bharthi et al. (2024) [58]	Canada	Yes	^c^ PGx-SParK clinical trial Mirror Image Trial of PGx testing implementation	Sample consisted of 90 young people aged (SD) 14 (2.5) years treated with Fluoxetine. Participants had an MDD or an OCD diagnosis.	Participants had DNA extracted from saliva samples and genotyped for *CYP2D6*, *CYP2C19*, *CYP2C9*, *CYP3A4* and *CYP3A5*.	Among the pharmacogenes tested, only the CYP2D6 activity score was associated with symptom improvement in young people treated with Fluoxetine. Higher CYP2D6 activity scores (faster drug clearance) were associated with a reduction in odds of reporting symptom improvement.
Gerlach et al. (2024) [59]	Canada	Yes	^c^ PGx-SParK clinical trial	A total of 214 individuals with a mean (SD) age of 12 (3.8) years participated in the trial, with *n* = 79 females Dose of Amphetamine and its duration were recorded.	Participants had DNA extracted from saliva samples and genotyped for *CYP2D6*. Symptom improvement (self-reported) and side effects.	Symptom improvement with *CYP2D6* depends on phenoconversion. Poor metabolisers had a 3x higher odds of reporting symptom improvement than intermediate metabolisers. Genotype predicted *CYP2D6* poor metaboliser phenotypes corrected for phenoconversion may assist in PGx-guided amphetamine prescribing in young people. No relationship between CYP2D6 phenotype and self-reported side effects.
Attia et al. (2024) [60]	Egypt	No	Retrospective case–control study	The study group consisted of 134 children aged (IQR) 8.0 years (5.0–11.0) with epilepsy and 124 age/gender matched healthy controls (7.0 years [5.3–8.8]). Children with epilepsy were categorised into those who were drug responsive (*n* = 67) and drug resistant (*n* = 67).	Genotyping of *rs2032582*, *rs717620*, *rs2273697*, *rs762551* and *rs3745274* polymorphisms	When compared to healthy controls, the study showed: Children with the ABCC2**rs717620* variant had a higher risk of seizures. Those with the ABCC2**rs2273697* and CYP1A2**rs762551* were considered to be protective against epilepsy susceptibility among drug-resistant patients when compared to drug-responsive ones. ABCB1**rs203258*2 and CYP2B6**rs3745274* polymorphisms did not confer a risk of epilepsy.
Gill et al. (2022) [61]	United States	Yes	Retrospective chart review study	The sample consisted of 27 patients who had PGx testing. All participants had problematic neurodevelopment/neurobehavioural disorders and aged: 11 (4) (mean [SD]) years old. About half of the study population was male.	Retrospective PGx testing for *CYP2D6* and *CYP2C19* pharmacogenes Review of PGx testing by multidisciplinary team	The study revealed that 40.8% and 59.3% participants had normal metaboliser status for CYP2C19 and CYP2D6, respectively. The most frequent disorders were ADHD (66.7%), anxiety (59.3%) and ASD (40.7%). About 66% of the participants had PGx actionable medication recommendations (start new therapy, switching medication, continuing current therapy or adding adjunct treatment). Approximately 25% of the sample had phenoconverted to a CYP2D6 poor metaboliser. Implementation of PGx testing could be facilitated by using a multidisciplinary model.
Zou et al. (2022) [62]	Canada	Yes	Observational study	In total, 104 participants, aged between 13 and 20 years, were recruited. Bipolar disorder (BD) cohort (*n* = 58): age (SD) 17.7 ± 0.2, female (66%) Healthy control cohort (*n* = 46): age (SD) 17.2 ± 0.3, female (50%) The BD cohort included those with BD-I, BD-II or BP-NOS.	Genotyping of *SOD2 rs4880* and *GPX3 rs3792797* SNPs Diffusion tensor imaging for whole-brain white matter analyses using fractional anisotropy and radial diffusivity.	The study showed that associations between SOD2 *rs4880* in white matter regions in both FA and RD were different between the BD group when compared to healthy controls. Within BD, differences were also revealed in the left corpus callosum. For *SOD2*, the G-allele had a higher antioxidant capacity than the A-allele. This suggests that the neuroprotective effects of the G-allele might be missing in the BD population and confer higher risk to oxidative stress. There was no association with the *GPX3 rs3792797* SNP.
Nussbaum et al. (2017) [63]	Romania	No	Observational study	Sample consisted of 210 patients (150 with psychosis and 60 patients with an ultra-high risk of developing psychosis) Median age (range) was 15.74 (13–20) years of age	Genotyping Assessments; PANSS, CGI-S/I and CGAS Spectroscopy	In those who benefited from PGx testing, assessment scores showed clinical improvement. Neuroimaging combined with PGx can be useful for managing psychosis in children and adolescents.
Nussbaum et al. (2016) [64]	Romania	No	Observational study	The sample had 87 participants with psychosis (median age: 15.78 ± 4 years). In this sample, *n* = 42 took treatment following PGx testing (*n* = 45 without). Sixty-five (65) were deemed to be of ultra-high risk for psychosis. In this sample, *n* = 32 took treatment following PGx testing (*n* = 33 without).	Genotyping of *CYP2D6* variants To evaluate psychopathology, the following assessments were used: (PANSS, CGI-S/I and CGAS) Neuroimaging	The study showed that PGx testing (combined with other methods) was useful in predicting treatment responses in children with psychosis and those at high risk. Neurobiology and neuroimaging markers can be a useful adjunct with PGx for predicting the clinical trajectory of psychosis in these patients.
Gassó et al. (2015) [65]	Spain	No	Observational study	All participants (*n* = 87) met the diagnostic criteria for OCD. Genetic data were available for 54 patients with a mean age (SD): 15.7 (2.1) years. Age of OCD onset was 13 years.	Genotyping of 262 polymorphisms in 35 candidate genes Clinical assessment using the CY-BOCS Neuroimaging methods	The study identified six polymorphisms associated with changes in white matter microstructure: *SLC1A1* (*rs3087879*), *SLC6A3* (*rs4975646*), *DRD3* (*rs3773679*), *NGFR* (*rs734194* and *rs2072446*) and *CDH9* (*rs6885387*). These polymorphisms are suggested to be involved in white matter changes in children with OCD.
Li et al. (2022) [66]	China	No	fNIRS observational study	Mean age of the study sample (*n* = 45) was 8.77 ± 1.16 years. However, 38 children completed the follow-up (mean age: 8.72 ± 1.16 years). Children were newly diagnosed with ADHD.	Detection of SNAP-25 gene *MnlI* polymorphisms ADHD questionnaire (SNAP-IV) was completed before and after 4 weeks of treatment with MPH fNIRS.	Lower SNAP-IV scores after MPH treatment were noted in those with the *T/T* genotype. The study results showed that a combination of fNIRs and SNAP-25 *MnlI* genotyping was useful in assessing treatment responses in children given MPH for ADHD.
Bruxel et al. (2015) [67]	Brazil	Yes	Observational PGx study	The total sample size was 655 (*n* = 523 with ADHD and 132 neurotypical controls). The PGx study included 172 children aged between 4 and 17 years with ADHD and treatment with Methylphenidate.	Genotyping of *LPHN3* (*rs6551665*, *rs1947275*, *rs6813183*, *rs1355368* and *rs734644*) polymorphisms The SNAP-IV was the primary outcome measure applied at baseline, 1st and 3rd months following treatment with Methylphenidate.	The key findings from the study showed that *LPHN3* polymorphisms are associated with ADHD susceptibility in children and adolescents. In particular, those with the *LPHN3* CGC haplotype have a higher risk. These polymorphisms also modulate the response to Methylphenidate treatment.
Poweleit et al. (2019) [68]	United States	Yes	Retrospective analysis of electronic medical data	The sample consisted of 352 participants who had underwent routine *CYP2C19* genotyping and 249 participants with additional genotyping of *HTR2A*, *SLC6A4* and *GRIK4* variants. All participants were less than 19 years old and were treated with Sertraline for anxiety and/or depressive disorder.	Retrospective review of electronic medical record data including *CYP2C19*, *HTR2A*, *SLC6A4* and *GRIK4* variant genotyping	The study demonstrated that *CYP2C19* variants (reduced functional alleles) and *HTR2A*, *SLC6A4* and *GRIK4* polymorphisms contribute to the treatment response and tolerability of Sertraline. Specifically, *CYP2C19* was associated with titration of Sertraline while other *HTR2A*, *SLC6A4* and *GRIK4* polymorphisms contributed to the maximum dose of Sertraline utilised, its titration and tolerability. The *HTR2A rs6313* polymorphism was associated with Sertraline dose response (*p* = 0.022).
Zai et al. (2023) [69]	^d^ United States	Yes ^Y^	Observational study	Paediatric (*n* = 37) patients with OCD aged 7–18 years Age, gender and total intracranial volume were covariates.	Assessment of two microsatellite markers (TAAA)n and (CA)n in MOG and its association with white matter volume measured using volumetric MRI Genotyping of *MOG* polymorphisms	The MOG (TAAA)n allele was shown to be associated with an increase in white matter volume (*p* = 0.018–0.028). The study findings indicated a role for MOG in the disruption of white matter and OCD and further confirm the notion that altered white matter is an endophenotype of OCD.
Ivashchenko et al. (2020) [70]	Russia	No	Observational study	Sample group had 36 adolescents (mean age [SD]: 14.83 years ± 1.84). All had an acute psychotic episode and were on an antipsychotic.	Assessment scales Genotyping of *CYP3A4*, *CYP3A5*, *CYP2D6*, *ABCB1*, *DRD2*, *DRD4* and *HTR2A* polymorphisms	Polymorphisms in *DRD2 rs1800497* and *ABCB1 3435C > T* were significant predictors of antipsychotic use. In summary, the study showed that these polymorphisms were associated with efficacy and safety of antipsychotics in young people with an acute psychotic episode.
Sági et al. (2021) [71]	^e^ Europe	No	Retrospective study	Study participants were aged between 1 and 18 years of age. All participants were treated for Acute Lymphoblastic Leukaemia. ATE *n* = 426, seizure subgroup *n* = 133, PRES *n* = 251 and case–control matched cohort *n* = 320	Retrospective review of clinical data Genotyping of 60 polymorphisms.	The study showed that gene polymorphisms *ABCB1*, *ABCG2* and *GSTP* are associated with chemotherapy-related CNS adverse events such as seizures and relapse.
Campagne et al. (2024) [72]	United States	No	Multicentre clinical trial	Children with brain tumours were less than 5 years of age Total number of children in the PGx analysis was 117 (mean age [range]: 1.97 years [0.09–4.94 years])	PK sampling Genotyping of genes associated with MTX/7OHMTX (22 SNPs from nine genes) Measurement of intracranial fluid volume	The study showed that *MTHFR*, *ABC* and *SLC* polymorphisms only had a modest influence on MTX metabolism but were not deemed to be clinically relevant.
Shilbayeh et al. (2024) [73]	Saudi Arabia	Yes	Prospective cohort study	The sample consisted of 89 children (mean age [SD]: 9.0 [4.1] years) with ASD treated with Risperidone	Markers of Risperidone exposure (plasma levels of Risperidone and its metabolite 9-OH-Risperidone) Exploratory genotyping	The study showed that: *rs78998153* in UGT2B17 was the most predominant PGx marker for Risperidone exposure. Human leukocyte antigen markers were also associated with Risperidone exposure parameters. PGx markers for Risperidone pharmacokinetics were not located in the genes typically associated with the Risperidone pathway.
Honeycutt et al. (2024) [74]	United States	Yes	Clinical trial	A total of 119 individuals (*n* = 66 genetic analysis; *n* = 48 PK analysis) aged 12–17 years of age participated in the trial. Participants had a first-degree relative with bipolar 1 disorder and had depression and anxiety symptoms. Treated with escitalopram.	Emergence of ADR in response to escitalopram Clinical assessment measures—TEASAP and PAERS scales. PGx assay at 8 weeks (or upon early discontinuation) *CYP2C19*, *CYP2D6*, *SLC6A4* and *HTR2A* genotypes were examined.	Reduced CYP2D6 metaboliser status was associated with an increased incidence of akathisia. *HTR2A A/A* or *A/G* genotypes were linked to a higher risk of self-harm. There were no adverse events associated with the CYP2C19 phenotype and *SLC6A4* genotype.
Aldrich et al. (2019) [75]	United States	Yes	Retrospective analysis of electronic medical data	Sample comprises 263 young people under 19 years who underwent CYP2C19 genotyping. All participants had anxiety and/or depressive disorder and were prescribed either escitalopram or citalopram.	Retrospective review of electronic medical record data, including routine *CYP2C19* genotyping	Those with a slow CYP2C19 metaboliser had poorer outcomes that those with faster metabolisers (*p* = 0.015) including behavioural activation (*p* = 0.029) and more weight gain (*p* = 0.018). More young people with slower metaboliser status discontinued treatment with escitalopram or citalopram than those with normal metabolisers (*p* = 0.007). The study highlighted the importance of CYP2C19 metaboliser status and treatment outcomes in young people given escitalopram or citalopram for the management of anxiety and/or depressive disorders.
Smith et al. (2017) [76]	United States	Yes	Case control study	All neonates had PDA. The sample was split into neonates who responded to Indomethacin (responders *n* = 96) and those who required surgical ligation (non-responders *n* = 52) for PDA. In the 96 responders, 53 were male, and in the 52 non-responders, 27 were male.	Genotyping of six polymorphisms *rs4918758*, *rs1799853*, *rs2253635*, *rs4086116*, *rs1505* and *rs2153628*) located in CYP2C9	Two CYP2C9 polymorphisms (*rs2153628* and *rs1799853*) were identified to be associated with Indomethacin response and treatment of PDA. Overall, in neonates with PDA, Indomethacin response can be influenced by *CYP2C9* polymorphisms. Further investigation of *CYP2C9* polymorphisms might help to mitigate the morbidity associated with PDA and Indomethacin toxicity.
Fan et al. (2021) [77]	Canada	Yes	Observational study	Prescribing rates per year in young people aged between 0 and 19 years Alberta drug dispensing data between 2012 and 2016.	Dispensing rates in drugs with PGx guidelines Among 61 drugs with PGx guidance, dispensing data were available for 57 drugs.	The study showed that between 1.1% and 45% of drug recipients had an actionable genotype, and dispensing rates increased over 5 years. Gene–drug pairs with the most impact were those associated with either *CYP2C19* or *CYP2D6* variants.
Rodriguez et al. (2021) [78]	Spain	No	Genome-wide methylation analysis study	Non-responders (*n* = 6): ➢ Mean age (SD): 13.3 (2.1) Responding (*n* = 6): ➢ Mean age (SD): 11.8 (3.1) All participants received CBT for the first time.	Gene methylation profiling Severity of OCD was captured using the CYBOCS completed at baseline and at 8 weeks of CBT.	The study identified two genes with significantly methylated positions and involved in the expression of non-coding RNAs: *PIWIL1* and *MIR886*. Overall, the study suggested that epigenetic modulation of non-coding RNA genes such as *PIWIL1* and *MIR886* could be a predictor of CBT response in children.
Gassó et al. (2017) [79]	Spain	No	Observational PGx study	Sample size (*n* = 83) consisted of participants aged (SD): 14.7 (1.7) years of age. All study participants received Fluoxetine treatment for psychiatric disorders (GAD, MDD or OCD).	Genotype analysis of polymorphisms located on TFBSs Clinical improvement following 12-week Fluoxetine use was assessed using the Children’s Depression Inventory scale.	This study showed that *rs11179002*, *rs60032326* and *rs34517220* polymorphisms were associated with higher clinical improvement following Fluoxetine treatment. Those with the rs34517220 polymorphism showed the greatest reduction in Children’s Depression Inventory scale scores.
Garfunkel et al. (2019) [80]	United States	Yes	Randomised controlled trial	All study participants had a diagnosis of ASD and gained weight while taking atypical antipsychotics were randomly assigned Metformin or placebo. Metformin cohort (*n* = 26): ➢ Mean (SD) age at consent: 12.7 (2.9) years, 23% female ➢ Mean (SD) Metformin dose: 1275mg (308) Placebo cohort (n = 27): ➢ Mean (SD) age at consent: 12.7 (2.5) years, 22% female ➢ Mean (SD) Metformin dose: 1346mg (203) In total, 53 participants had DNA samples that were analysed using a linear, mixed model analysis.	Genotyping of five genes involved in metformin response Mean change in BMI z-score (primary outcome measure)	Of the polymorphisms investigated, *ATM* and *OCT1* genotypes had significant effects on BMI z-scores during 16 weeks of Metformin treatment. Overall, the study showed that genetic variation can be useful in predicting Metformin responses in children with ASD on atypical antipsychotics.
Chidambaran et al. (2015) [81]	United States	Yes	Prospective observational study	Sample consisted of 88 young people aged 11–18 years of age (mean [SD]; 14.59 [1.89]). All participants had a diagnosis of idiopathic scoliosis and/or kyphosis surgery for spinal fusion between 2008 and 2013.	Genotyping for *OPRM1* polymorphism Primary outcome measure was MIRD that occurred between 2 to 48 h post-surgery.	Participants with the *OPRM1 A118G AA* genotype had a higher risk of MIRD (*p* = 0.030). Those with the G allele, i.e., GG or AG, had a lower sensitivity to morphine and thus a lower risk of MIRD. Overall, the study indicated that the *OPRM1 A118G* polymorphism has a role in modulating the depressant and analgesic effects of morphine in young people undergoing surgery for spinal fusion.
Sadhasivam et al. (2015) [82]	United States	Yes	Observational study	Patients were aged between 6 and 15 (*n* = 263) years and had outpatient tonsillectomy	Genotyping *ABCB1* variants Metrics for intravenous morphine effectiveness	The study showed that those carrying the GG and GA alleles of the *ABCB1* polymorphism (*rs9282564*) had (I) increased risk of respiratory depression and (II) prolonged hospitalisation. An additional copy of the G allele increased the likelihood of hospitalisation x4.7-fold following surgery.
Vande Voort et al. (2022) [83]	United States	Yes	Randomised controlled trial	The total sample was 176, of which 84 had PGx tests available at baseline (GENE arm) and 92 had PGx testing available at week 8 visit (TAU arm) Participants were aged 13–18 years old with major depressive disorder.	Outcome measures (assessment scales) with Child Psychiatrist oversight The PGx panel test for the following 8 pharmacogenes: *CYP2D6, CYP2C19, CYP2C9, CYP3A4, CYP2B6, CYP1A2, SLC6A4* and *HTR2A* Side effects and symptom improvement were assessed at 4 and 8 weeks, and 6 months	There were no statistically significant differences in the outcome measures between the GENE and the TAU arm at either 8 weeks or 6 months. Symptom improvement, side effects or satisfaction did not differ when PGx testing was available at either baseline or after 8 weeks, suggesting that PGx may not have any direct clinical impact as evaluated by the different outcome measures.
Nooraeen et al. (2024) [84]	United States	Yes	Randomised Controlled Clinical Trial—post hoc analysis	Sample: 196 patients were screened, and 170 adolescents were included in secondary analysis. Patients were grouped according to whether they had a low (*n* = 60), medium (*n* = 83) or high risk (*n* = 27) of gene–drug interactions. All had moderate to severe depression and were aged 13 to 18 years of age.	PGx testing Symptom improvement and side effects at baseline, week 4, week 8 and 6 months Outcome measures (assessment scales) with Child Psychiatrist oversight	While symptoms of depression did not change, this study showed that those patients taking medications with a high risk of gene–drug interactions were more likely to have side effects when compared to patients with low/medium gene–drug interactions at week 8 (*p* = 0.001). Examining specific medication–gene pairs could prove more useful than choosing a medication in the “low gene–drug interaction” group. Overall, PGx could be useful for minimising medication side effects rather than focusing solely on efficacy.
Liko et al. (2021) [85]	United States	Yes	Cross-sectional study	Survey was sent to 1358 healthcare providers, of which 19.2% responded (*n* = 261). Healthcare providers consisted of 37 different specialties (general paediatrics—largest specialty [15%]). Among respondents’ physicians (*n* = 173, 66.3%), pharmacists (*n* = 42, 16.1%), nurse practitioner (*n* = 31, 11.9%), physician assistant (*n* = 13, 5.0%) and another provider (*n* = 2, 0.8%)	A survey consisting of 26 items The survey assessed the following related to PGx testing: knowledge (domain 1), attitudes (domain 2), perceptions (domain 3) and experiences (domain 4).	The study showed that of the 261 respondents, 71.3% were “slightly or not at all familiar” with PGx, even though more than half had some prior knowledge or training. About 70% of respondents rated PGx testing as “either moderately, very or extremely useful” Paediatric specialties rating PGx testing as most clinically useful were as follows: haematology/oncology (63.8%), anaesthesiology/pain and surgery (59.5%) and psychiatry (46.7%). Unfamiliarity with PGx testing (78.2%) followed by cost/insurance coverage (62.8%), application to clinical practice (62.5%) and clinical value of PGx testing (60.2%) were rated as the most common challenges.
Cohn et al. (2021) [86]	Canada	No	Cohort study consisting of two patient cohorts: 1, Point-of-care (reactive—based on targeted drug–guided testing) 2, Pre-emptive (whole-genome sequencing–guide testing)	The study group consisted of 172 children (mean [SD] age: 8.5 [5.6] years, 108 boys and 64 girls) Point-of-care cohort (*n* = 57): ➢ Mean (SD) age: 10.3 (5.5) years, 32 boys and 25 girls ➢ Medical conditions varied in this cohort Pre-emptive cohort (*n* = 115): ➢ Mean (SD) age: 7.6 (5.4) years, 76 boys and 39 girls ➢ Patients had cardiac disease	In both cohorts, the main outcome measure was the number of patients with a PGx test that resulted in a deviation from standard treatment regimens. For the point-of-care cohort, 57 patients were referred to the PGx clinic for drug guidance based on *CYP2C19, CYP2C9*, *CYP2D6*, *CYP3A5*, *VKORC1* and *TPMT* gene–drug information concerning cardiovascular medications, proton pump inhibitors and psychiatric agents. For the pre-emptive cohort, 115 patients received exploratory WGS, and data from six pharmacogenes (*CYP2C19, CYP2C9*, *CYP2D6*, *CYP3A5*, *VKORC1* and *TPMT*) were extracted.	The PGx testing in 172 patients for 6 pharmacogenes resulted in a modification in standard treatment in 36.8% and 80% of paediatric patients in the point-of-care and preemptive cohorts, respectively. In the point-of-care cohort, the median number of genes examined per patient was 2, and *CYP2C19* was the most frequently examined (n = 52). About 50% also had 1 variant in 6 pharmacogenes examined. Both standard and modified treatment regimens were clinically actionable. About 73% of patients were from cardiology services and highlighting the benefits of identifying patients from high-risk services who could be at higher risk of ADRs with standard dosing regimens.
Davis et al. (2021) [87]	United States	Yes	Open-label CBD study	The sample consisted of 169 patients, of whom 54.5% were paediatric Fifty (50) % were female All patients had treatment-resistant epilepsy	Genotyping for CBD response. Assessment of CBD response, safety and tolerability.	The study indicated that PGx variation is linked with CBD responses in treatment-resistant epilepsy. Genetic variations in cytochrome P450 enzymes were associated with a lower response to CBD. Variations in pharmacogenes revealed associations between cholesterol and glutathione metabolism. The study also showed interactions between medications such as statins and acetaminophen.
Gota et al. (2016) [88]	India	No	Observational study	Median age (range) of the sample (*n* = 35, 26 male and 9 female) was 5 years (range 1–13) All patients were receiving 13-cis retinoic acid for the treatment of neuroblastoma.	Genotyping of *UGT2B7*, *CYP3A5, CYP3A7* and *CYP2C8* polymorphisms	The study showed that genetic variation in *CYP* and *UGT* polymorphisms does not modify the metabolism of 13-cis retinoic acid in patients being treated for neuroblastoma
Hallik et al. (2022) [89]	Estonia	No	Clinical trial	Median age range was 30.9 weeks. Of the study participants, 26/28 were involved in the PGx study.	Genotyping of SNPs: β1, β2 adrenoceptor (AR) and Gs protein α-subunit gene (*GNAS*) Assessment of heart rate parameters	The study showed that *β1-AR Arg389Gly* and *GNAS c.393C > T* polymorphisms were associated with the haemodynamic response to dobutamine in severely ill neonates.
Johnson et al. (2021) [90]	^f^ Consortium	Yes ^µ^	Multicentre genetic study	Four patients (age at evaluation: 20 years, 10 years, 11 years and 34 years old) with juvenile ALS enrolled in the study between 2016 and 2021 The age of onset of juvenile ALS ranged between 5 and 15 years. Sample: juvenile ALS *n* = 63 and adult ALS *n* = 6258 screened for *SPTLC1* = total 66 and 6258	Clinical history and assessment Next-generation sequencing Mitochondrial assays and sphingolipid measurements	The study showed that de novo variants in the *SPTLC1* gene are associated with juvenile ALS. The overall finding suggested that patients with juvenile ALS should be screened for the variants in the *SPTLC1* gene.

Notes: ^¥^ All study participants were Saudi. ^Y^ Described in Barr et al. Linkage study of catechol-O-methyltransferase and attention-deficit hyperactivity disorder. Am. J. Med. Genet. 1999 Dec 15;88(6):710-3. ^µ^ Reported for patients 2, 3 and 4. ^a^ The study was developed by the IGNITE working group. Data were used from the 16 sites, consisting of 12 academic medical centres and 4 community hospitals or clinical systems. ^b^ EpiPGX Consortium. ^c^ Study participants were enrolled as part of the Pharmacogenetic-Supported Prescribing in Kids (PGx-SParK) trial. ^d^ Participants were recruited for this study from the Paediatric Anxiety Disorders Clinic, Wayne State University. ^e^ The study included participants from Hungary, Austria, the Czech Republic and the Nordic Society for Paediatric Haematology and Oncology. ^f^ Consortium members are the FALS Sequencing Consortium, American Genome Centre, International Amyotrophic Lateral Sclerosis (ALS) Genomics Consortium and the Italian ALS Genetic (ITALSGEN) consortium.

**Table 2 genes-16-01055-t002:** Frequency of themes.

Theme	Sub-theme	Count *
Implications of non-CYP450 polymorphisms	Disorder-Specific Associations	19
	Treatment Response and Efficacy	3
Paediatric CYP450 PGx	Treatment Response and Efficacy	5
	CYP450 Substrates and Gene–drug Pairs	3
	Disorder-Specific Associations	3
Genetic predictors of response		8
Insights for implementation and future research		7
Phenoconversion		4

* Count represents the number of articles included in the theme.

## Data Availability

The data extracted and used in this systematic review were derived from databases that are easily accessible in the public domain.

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
