# Peer review of "A Comprehensive Analysis Examining the Role of Genetic Influences on Psychotropic Medication Response in Children"

_genes, 2025, doi:10.3390/genes16091055_

Round 1
Reviewer 1 Report
Comments and Suggestions for Authors
The manuscript by Singh et al presents a systematic review of 50 studies investigating genetic influences on psychotropic medication response in children and adolescents. The authors follow PRISMA methodology, apply the Downs & Black and CHEERS checklists, and identify five thematic categories: (1) non-CYP450 polymorphisms in pediatric disorders, (2) pediatric cytochrome P450 pharmacogenetics, (3) genetic predictors of treatment response, (4) insights for implementation and future research, and (5) phenoconversion. The topic is timely and clinically relevant, given the paucity of pediatric-specific pharmacogenetic (PGx) evidence.
Major Concerns
Critical Synthesis vs. Description
The manuscript focuses on describing the themes of the studies, while doing little to synthesize those themes into meaningful and impactful findings. The manuscript instead focuses on a study-by-study breakdown of genes and polymorphisms. The manuscript would benefit from an integrative approach where a theme is identified and then placed in the broader context of existing literature and reporting studies, rather than for each study.
Developmental Considerations
Pediatric pharmacogenetics differs fundamentally from adult populations due to the ontogeny of drug-metabolizing enzymes and developmental neurobiology. Although acknowledged, this is not sufficiently integrated into the analysis. The review would benefit from a more detailed discussion of how the developmental stage modifies pharmacogenetic relevance.
Study Heterogeneity
The review includes 50 studies that range from case reports to observational studies, randomized controlled trials, and population-level studies. While this is comprehensive, little is done to evaluate the relative strength of the evidence reported by each study. For example, does the evidence for ABCB1 from Kalla et al (2023) have the same level of impact as the one by Sadhasivam et al (2015)? An acknowledgement of this heterogeneity and subsequent weighting of the themes might help.
Quality Appraisal Interpretation
The Downs & Black and CHEERS scores are reported, but their significance is not critically explored. What does an average Downs & Black score of 17.7/27 imply for confidence in the conclusions? What does a CHEERS score of 33% mean for real-world implementation? The discussion should more directly address whether current evidence is strong enough to influence guidelines.
Minor Concerns
Writing Style:
Some sections are dense, with long lists of genes and polymorphisms. A more narrative synthesis highlighting only the most clinically relevant findings would improve readability.
Figures and Tables:
- The Prisma Flow chart (Figure 1) can be clarified. In box 1 it shows the total number of search results ( n = 769), but in the second one, it shows the number that were excluded. The authors can use something like (https://estech.shinyapps.io/prisma_flowdiagram/) to generate a diagram.
- The thematic frequency tree can be better visualized as a table.
Terminology:
Ensure consistent use of terms (e.g., pharmacogenetics vs. pharmacogenomics vs.
PGx).
Formatting:
Minor inconsistencies in references and spacing should be corrected.
Methods Transparency:
More detail on thematic analysis (coding framework, inter-rater reliability) would enhance reproducibility.
Strengths
- Timely and clinically important review in an under-researched area.
- Comprehensive search strategy across multiple databases with PRISMA adherence.
- Inclusion of blinded dual review and consensus approach.
- Broad coverage of pediatric psychopharmacogenetics, capturing both CYP450 and non-CYP450 variants.
- Clinical orientation with discussion of CYP2D6/CYP2C19 metabolism, autism, ADHD, epilepsy, and psychosis.

Author Response
Please find attached my responses to Reviewer 1.

Reviewer 2 Report
Comments and Suggestions for Authors
In their mansucript "a comprehensive analysis examining the role of genetic influences on psychotropic medication response in children" the authors have undertaken a systematic review and thematic analysis of an area that remains critically underexplored: the genetic underpinnings of psychotropic medication responses in children. The scale of the work is remarkable, synthesizing findings from 50 studies with sample sizes ranging from extremely small cohorts to population-level data encompassing nearly three million individuals. The careful application of PRISMA guidelines, together with rigorous quality and economic appraisals using the Downs and Black and CHEERS checklists, highlights the significant methodological investment and attention to transparency that went into this work.
The study is especially valuable because it addresses a highly vulnerable group—children with mental health and neurodevelopmental conditions—for whom treatment decisions carry long-term developmental implications. Whereas pharmacogenetic research in adults is relatively well established, the pediatric field is fragmented, methodologically heterogeneous, and underrepresented in clinical guidelines. By systematically reviewing and thematically organizing the evidence, this manuscript provides a much-needed resource for clinicians, researchers, and policymakers working at the intersection of child psychiatry, pharmacogenomics, and personalized medicine.
The thematic analysis, which identified five key domains (non-CYP450 polymorphisms, pediatric cytochrome P450 pharmacogenetics, genetic predictors of response, implementation and future research, and phenoconversion), reflects a thorough engagement with the available literature. Particularly noteworthy is the attention to non-CYP450 polymorphisms—an area often overshadowed by CYP450-focused studies—demonstrating the potential of novel genetic markers to inform treatment pathways in children with autism, ADHD, epilepsy, and other neurodevelopmental disorders.
While the manuscript succeeds in mapping the field comprehensively, one area that could be strengthened further is the discussion of practical clinical implications. The review clearly shows the promise of pharmacogenetics in pediatric psychiatry, but the translation of these findings into real-world clinical pathways could be more explicitly articulated. For example, how might clinicians prioritize genetic testing in routine practice? Should PGx be considered primarily for children with treatment-resistant conditions, or could it also be valuable pre-emptively to avoid adverse drug reactions? Similarly, while the absence of health economic models is rightly highlighted, the authors could go further in suggesting pragmatic approaches for embedding PGx into pediatric care despite current cost and coverage barriers. Addressing these questions would make the work even more actionable for the multidisciplinary teams who ultimately need to apply these insights.
In sum a minsor revision focusing on a slightly stronger emphasis on clinical translation is recommended.
Author Response
Please find attached my responses to Reviewer 2.

Round 2
Reviewer 1 Report
Comments and Suggestions for Authors
The authors have clearly and comprehensively addressed the concerns. I believe that this is okay to go ahead with publication.